# Comparative Structure-Property Characterization of Poly(3-Hydroxybutyrate-Co-3-Hydroxyvalerate)s Films under Hydrolytic and Enzymatic Degradation: Finding a Transition Point in 3-Hydroxyvalerate Content

**DOI:** 10.3390/polym12030728

**Published:** 2020-03-24

**Authors:** Vsevolod A. Zhuikov, Yuliya V. Zhuikova, Tatiana K. Makhina, Vera L. Myshkina, Alexey Rusakov, Alexey Useinov, Vera V. Voinova, Garina A. Bonartseva, Alexandr A. Berlin, Anton P. Bonartsev, Alexey L. Iordanskii

**Affiliations:** 1Research Center of Biotechnology of the Russian Academy of Sciences 33, Bld. 2 Leninsky Ave, 119071 Moscow, Russia; zhuikova.uv@gmail.com (Y.V.Z.); tat.makhina@gmail.com (T.K.M.); v.l.myshkina@gmail.com (V.L.M.); bonar@inbi.ras.ru (G.A.B.); ant_bonar@mail.ru (A.P.B.); 2Federal State Budgetary Institution “Technological Institute for Superhard and Novel Carbon Materials”, 7a Tsentralnaya Street, Troitsk, 108840 Moscow, Russia; rusakov.alexey@gmail.com (A.R.); useinov@mail.ru (A.U.); 3Faculty of Biology, M.V. Lomonosov Moscow State University, Leninskie Gory 1-12, 119234 Moscow, Russia; veravoinova@mail.ru; 4Research Center of Chemical Physics the Russian Academy of Sciences, Kosygin str. 4, 119991 Moscow, Russia; berlin@chph.ras.ru (A.A.B.); aljordan08@gmail.com (A.L.I.)

**Keywords:** poly(3-hydroxybutyrate), poly(3-hydroxybutyrate-co-3-hydroxyvalerate), poly(3-hydroxybutyrate-co-4-methyl-3-hydroxyvalerate), biodegradation, hydrolysis, pancreatic lipase, mechanical behavior

## Abstract

The hydrolytic and enzymatic degradation of polymer films of poly(3-hydroxybutyrate) (PHB) of different molecular mass and its copolymers with 3-hydroxyvalerate (PHBV) of different 3-hydroxyvalerate (3-HV) content and molecular mass, 3-hydroxy-4-methylvalerate (PHB4MV), and polyethylene glycol (PHBV-PEG) produced by the *Azotobacter chroococcum 7B* by controlled biosynthesis technique were studied under in vitro model conditions. The changes in the physicochemical properties of the polymers during their in vitro degradation in the pancreatic lipase solution and in phosphate-buffered saline for a long time (183 days) were investigated using different analytical techniques. A mathematical model was used to analyze the kinetics of hydrolytic degradation of poly(3-hydroxyaklannoate)s by not autocatalytic and autocatalytic hydrolysis mechanisms. It was also shown that the degree of crystallinity of some polymers changes differently during degradation in vitro. The total mass of the films decreased slightly up to 8–9% (for the high-molecular weight PHBV with the 3-HV content 17.6% and 9%), in contrast to the copolymer molecular mass, the decrease of which reached 80%. The contact angle for all copolymers after the enzymatic degradation decreased by an average value of 23% compared to 17% after the hydrolytic degradation. Young’s modulus increased up to 2-fold. It was shown that the effect of autocatalysis was observed during enzymatic degradation, while autocatalysis was not available during hydrolytic degradation. During hydrolytic and enzymatic degradation in vitro, it was found that PHBV, containing 5.7–5.9 mol.% 3-HV and having about 50% crystallinity degree, presents critical content, beyond which the structural and mechanical properties of the copolymer have essentially changed. The obtained results could be applicable to biomedical polymer systems and food packaging materials.

## 1. Introduction

Poly (3-hydroxybutyrate) (PHB), the main polymer of the polyhydroxyalkanoates family (PHA), is the most well-known microbiological polyester, which is a promising alternative to biodegradable synthetic thermoplastics [1,2,3] and other biocompatible polymers [4,5,6,7,8,9,10,11]. PHA are obtained microbiologically. This production method allows varying the physicochemical properties of polymers of this type over a wide range [12]. Because PHB has the ability to biodegrade and has high biocompatibility, it is widely used for biomedical applications in regenerative medicine and tissue engineering [13,14,15,16,17,18,19,20,21,22] and medicine forms [23,24,25,26]. It is able to create composites with synthetic polymers, inorganic materials [19,27,28,29] and also as a new environmentally friendly material, including application as a material for the packaging industry [15,24,30,31].

However, PHB has a number of disadvantages: fragility, lack of hydrophilicity, etc. In order to improve the physicochemical properties, copolymers of PHB with other polyhydroxyalkanoates are made. The copolymer poly(3-hydroxybutyrate-co-3-hydroxyvalerate) is made to compensate disadvantages of PHB. Incorporation of HV into the PHB homopolymer chain considerably improves the physicochemical properties, such as the melting temperature. The copolymer PHBV is more plastic, extensible, and resilient due to a decrease in the value of Young’s modulus with an increase in the HV molar fraction in PHB–HV polymer chain Reference [32] The copolymerization of PHB with polyethylene glycol (PEG) increases water permeability and solubility due to the hygroscopicity of PEG. In addition, biocompatibility and hydrophilicity are improved compared to a homopolymer [15].

It is well known that the biodegradation of PHB both in living systems and in the environment occurs through enzymatic and non-enzymatic processes that occur simultaneously in natural conditions, compared to other biodegradable polymers (e.g., poly(lactic-co-glycolic) acid [33]). PHB is considered to be moderately resistant to in vitro degradation, as well as to biodegradation in animal tissues. The degradation rate is influenced by polymer characteristics, such as chemical composition, crystallinity, morphology, and molecular mass [34,35].

Therefore, to develop novel medical devices and packing materials based on PHA and its copolymers, it is necessary to know how the physicochemical properties of these polymers change during their degradation. In order to understand what changes, occur with polymers in the human body during degradation, it is necessary to study the kinetics of the change in the basic physicochemical properties when they degrade in vitro under conditions simulating the internal environment of the organism. Studies of degradation of PHA, especially on long periods, are rare [36]. Thus, the purpose of this study is to obtain and compare the kinetic curves of the long-term degradation of PHB and its copolymers. Considerable attention is paid to the change in the molecular mass and degree of crystallinity of polymer films. In addition, at present in the literature, there are no exact data on the effect of the monomer composition of the copolymer on the polymer decomposition process. There is fragmentary evidence that the inclusion of > 10% 3-hydroxyvalerate (3-HV) in the composition of the copolymer leads to a change in the structure of the crystalline component of the polymer [37].

An understanding of the processes of hydrolytic and enzymatic degradation of PHAs is very important for the development of a new biodegradable and harmless packaging for meat, fish, dairy, and vegetable food products [38,39,40,41].

Therefore, the goals of this work are to trace the changes in the physicochemical properties of the copolymers and to search for the value of the included hydroxyvalerate in the PHV chain, which has a key effect on the degradation mechanisms of the product.

## 2. Materials and Methods

### 2.1. Production of Films from PHAs

To study the in vitro biodegradation, a series of films 50 ± 10 μm in thickness and 90 mm in diameter was made from PHB and its copolymers (Table 1) obtained by controlled biosynthesis using producing strain Azotobacter chroococcum (Appendix A). The polymer (~400 mg) was dissolved in chloroform (35 mL) overnight at room temperature. Films were prepared by casting from a chloroform solution to the bottom of Petri dishes previously degreased. The films were dried until the solvent was completely removed for at least 72 h at room temperature. Then, the plates were cut from the resulting films with dimensions of 30 mm × 10 mm.

### 2.2. Determination of MM

The molecular mass (MM) of PHB and its copolymers were determined by gel filtration chromatography (GPC). Chloroform with the addition of 3% v/v methanol was used as a solvent. The elution rate was 1 mL/min. Used as the detector was a Waters 2414 differential refractive index detector and a UV detector and a waters 1525 pump. The sample was 100 μL with a concentration of 5 mg/mL. Four Waters styragel columns (Waters, Milford, MA, USA) (Styragel HT 6E, 4.6 mm × 300 mm) were used. Calibration was carried out using polystyrene reference samples having a narrow distribution [31]. The data obtained by GPC were correlated with viscometric data estimated by viscosimetry [40]. The viscosity was measured in a 30 °C solution of chloroform in a Ubbelohde viscometer. Molecular mass was calculated using the Mark-Hauwink-Kuhn equation. The 6 specimens per sample were analyzed (Appendix A).

A mathematical description for the not autocatalytic and autocatalytic degradation of aliphatic polyesters mechanisms was proposed in Reference [42]. Assuming that the degree of degradation is low, the authors proposed the following kinetic dependence based on the mean molecular mass of polymers (1):1/MM = 1/MM_0_ + kt,(1)
where MM and MM_0_ are the mean molecular mass of the polymer component at time t and at the initial time, respectively, and k is the rate constant. An equation that took autocatalysis into account, which is the consequence of the appearance of the terminal groups of carboxylic acids, was also proposed. The following equation can describe this process (2):
ln(MM) = –kt + lnMM_0_.(2)

### 2.3. Determination of Crystallinity of a Polymer

The crystallinity of PHA were measured by DSC (DSC 204 F1 Phoenix, Netzsch, Germany). The samples were heated from 25 to 220 °C. at a heating rate of 10 K/min under an argon atmosphere. The crystallinity of the PHA structure (Xc) can be calculated as follows (3):
Xc = (ΔHm/ΔH_0_m(PHB)) × 100%,(3)
where ΔHm is the enthalpy change caused by the melting of the test sample, respectively, and ΔH_0_m (PHB) is the theoretical value for the thermodynamic enthalpy of melting that would have been obtained for 100%-crystalline samples of the PHB (146.6 J/g) [43]. All calculations were carried out for the second heating cycles (Appendix A). The 6 specimens per polymer sample were analyzed.

### 2.4. Exploration of Mechanical Properties

The mechanical properties of the films were studied using the nanoindentation method in accordance with ISO 14577. The measurements were carried out using a NanoScan-4D scanning nano-hardness tester (TISNCM, Troitsk, Moscow, Russia). Nanoindentation was performed on the smooth side of the films. Films with dimensions of 2 mm × 2 mm were fixed with phenyl salicylate. The load was carried out in a linear mode, the peak load on the sample was 5 mN. The load time was equal to the unloading time and was 30 s. The peak load was maintained for 5 s. The average penetration depth into the sample was not more than 10% of the film thickness (Appendix A). The 6 specimens per polymer sample were analyzed.

### 2.5. Contact Angle Measurement

The hydrophilicity of the polymer surfaces was evaluated by measuring the contact angle between the drops of water and the “smooth” surface of the samples using the Contact Angle Meter 110 VAC (Cole-Parmer, Vernon Hills, IL, USA). For this purpose, a drop of distilled water (10 μL) was applied to the surface of the films, and then the contact angle of wetting was measured. The values were averaged over the corners obtained from 10 drops per film (Appendix A). The 6 films were analyzed for each polymer sample.

### 2.6. In Vitro Degradation Experiment

The study of the degradation of PHA films was carried out as follows. The plates were incubated in 15 mL of phosphate-buffered saline (PBS) and in 15 mL of a solution of porcine pancreatic lipase in PBS (Sigma L3126) having a pH of 7.4 at 37 °C in a thermostat for 183 days. The lipase concentration in the phosphate buffer solution was 0.25 mg/mL. The concentration of pancreatic lipase was selected based on earlier obtained experimental data (Appendix A). The pH was monitored with an Orion 420 + pH meter (Thermo Fisher Scientific, Waltham, MA, USA). To assess the changes in the mass of the polymer plates, the test plates were withdrawn from the lipase solution after 1 week, 1 month, 3 months, and 6 months, dried, and weighed on a scale. The average mass of the plates was 15–25 mg. The change in the mass of the plates during the degradation was determined gravimetrically on the AL-64 scales (Max = 60 g, d = 0.1 mg, ACCULAB, Bohemia, NY USA). To prevent bacterial contribution to the degradation of polymers, sodium azide (2 g/L) was added to the buffer solution, and the buffer solution was replaced twice per week [37,44].

### 2.7. Statistical Analysis

For statistical analysis a one-way ANOVA was applied. In the tables and figures, the data were presented as mean values and standard deviation (M ± SD) at a significance level of P < 0.05.

## 3. Results

### 3.1. The Decrease in Mass of PHA Films

Biodegradation of PHA occurs as a result of a combination of hydrolytic and enzymatic degradation. This leads to a change in the mass of the samples and their physical and chemical properties [1,2,12,44,45]. The analysis of the degradation kinetic curves (Figure 1) showed that during the first week, the mass of all the samples under study was decreased.

All the investigated PHA films did not show the visual erosion during the entire degradation process. Subsequently, the dry mass of the PHA films placed in the lipase solution did not demonstrate significant changes even after 180 days. The mass of all the samples did not decrease more than to 92% of the initial mass, indicating a rather slow degradation in the lipase solution. The greatest loss (~ 8–9%) of the mass was observed in the PHBV 17.6% 1190 and PHBV 9% 1010.

### 3.2. Changes in Molecular Mass (MM)

MM is one of the most significant parameters of polymer degradation. The value of the molecular mass of natural PHAs can be specified by the controlled bacterial biosynthesis and polymer chemical processing [46]. When studying the biopolymers degradation, considerable attention is paid to the MM, because it has a great effect on the other physicochemical parameters of biopolymers [47]. In addition, the MM values are the most indicative degradation parameter. It is extremely sensitive to the polymer backbone destruction. It was found that with a slight change in the mass of the samples, MM underwent more significant changes. The special features of MM changes during biodegradation make it possible to understand the type of its mechanism: not autocatalytic or autocatalytic [42,48].

Within 6 months of incubation in PBS and in lipase, the greatest loss of MM is observed in sample PHB 1095 that was up to 80% of the initial MM (Figure 2). For the other polymer samples, a decrease in MM was also observed, but no dependence on the molar content of HV was detected. Only two polymers from this group in the first week increased their MM–PHBV 17.6% 1190 and PHBV 5.9% 819.

To analyze the curves of the MM decrease during the degradation, the decomposition models of partially crystalline polymers were applied [42,49]. To evaluate the applicability of the specific model, the curves of degradation were calculated via the statistical correlation coefficients. For this object, the plots of 1/MM and ln(MM) versus degradation time, reflecting not autocatalytic (not autocatalytic for enzymatic degradation) and autocatalytic mechanism, respectively (Figure 3А,B).

Figure 3A shows that the analyzed curves are aligned with the increase in the molar content of HV in the PHBV chain (from homopolymer (line #1) to copolymer with 17.6 mol% of HV (line #5).

By presenting the tangent of the slope of the curves as a function of the HV content, the graph of the slope angle dependence for the not autocatalytic model was obtained (Figure 4).

The intersection point for two lines corresponds to 5.9 mol. % HV in PHBV. These data confirm that, at this molar content of PHBV, the polymer structure and corresponding mechanical properties could be modified.

In accordance with the described models, correlation coefficients were determined for each curve (Table 2). In Table 2, it is shown the correlation coefficients for hydrolytic and enzymatically catalytic models.

The correlation coefficients of the not-autocatalic model were quite high (>90%) for the both hydrolytic and enzymatic degradation, while the correlation coefficients of the autocatalic model were higher for the enzymatic degradation with the exception of polymers PHB 1095 and PHBV 2.5% 768.

### 3.3. Degree of Crystallinity

The crystallinity degree of PHBV was calculated on the basis of the melting heat for the completely crystalline PHB (146.6 J/g) [43]. Comparing the graphs, Figure 5A,B, you can see that in the initial period of time the changes in crystallinity in different solutions are similar. In both cases, in the period up to 1-month, strong changes in the degree of crystallinity were observed.

It was shown that the degree of crystallinity for the copolymers is less than for the sample of PHB 1095. Thus, if the degree of crystallinity for intact PHB 1095 before incubation was 63%, the degree of crystallinity of the PHBV copolymer with the highest molar HV content (17.6 mol. %) at the initial moment was 34%. It is almost half that of PHB 1095. It can also be noted that the degree of crystallinity decreases with increasing molar content of 3-hydroxyvalerate chains.

With further degradation of the polymers for 6 months, incubation of PHA films in PBS resulted in a wave-shape change in the degree of crystallinity (Figure 5A). However, the degree of crystallinity changed differently when incubated in a solution of phosphate buffer with pancreatic lipase. The wave-shape change in crystallinity did not occur (Figure 5B), and its values were observed only in PHB 1095 and PHBV 2.5% 768. It is also important to note that, unlike hydrolytic degradation, there was a clear tendency (Figure 5B) to decrease the degree of crystallinity as the function of the molar content of 3-HV in the homopolymer chain. In addition, the slope of the curves (the values of tgα in Figure 5B) also changed. During degradation, the crystallinity of PHB 1095 increased over 6 months, however, the crystallinity of the PHBV 17.6% 1090 decreased through 6 months. So, it is possible to calculate the molar content of 3-HV in the PHBV chain, which begins to affect the structural and mechanical properties of the entire polymer, which leads to a different character of polymer decomposition (Figure 6).

Figure 6 shows that after incubation in a lipase solution by the 6th month of degradation, the degree of crystallinity was decreased if the percentage of 3-HV in the PHBV chain exceeded 5.7%. This is very close to the value determined earlier in at the analysis of MM changes for the copolymer with HV content equals to 5.9%.

### 3.4. The Change in Mechanical Properties of PHA Films

Changes in the mechanical properties of the copolymers as a result of degradation were measured by the nanoindentation method (Figure 7). It should be noted that Young’s modulus of the homopolymer (2.2 ± 0.06 GPa) before degradation was higher than that of the copolymers (~1 GPa). This is due to the fact that the copolymers have a lower degree of crystallinity and, consequently, stiffness.

During the first week, Young’s modulus of all polymers was sharply increased (Figure 7). The largest value of Young’s modulus was observed for PHB 1095 kDa. Over a month, Young’s modulus of the homopolymer increased to 4.7 ± 0.1 GPa and remained at this level throughout the entire experiment (6 months). Young’s modulus of the other copolymers also increased during the first week and did not change significantly in the future. However, the stiffness of the copolymers is still less than that of the homopolymer, for comparison, Young’s modulus of the PHBV 17.6% 1190 for the 6-month degradation was 2.2 ± 0.07 GPa, which was half that of Young’s modulus of homopolymers.

### 3.5. The Change in Hydrophobicity of PHA Films

The balance between hydrophobicity and hydrophilicity of the surface is one of the main characteristics indicating biocompatibility of the surface. Biocompatibility is one of the most important properties of polymers that can be used in medicine, so the degree of hydrophilicity of the surface of the polymer affects the growth of cells [3].

During the biodegradation, the contact angle between the standard drop of water and the surface of the polymer film decreased, which indicates that the hydrophobicity degree for the copolymers decreases (Figure 8).

In general, during biodegradation, the contact angle for all copolymers contacted with lipase decreased by an average value of 23% compared to 17% for hydrolytic degradation (the differences are statistically significant, p ≤ 0.01).

## 4. Discussion

Thanks to their unique properties, PHAs are the most promising polymers for use in various fields, such as ecology, biomedicine, and packaging. PHAs are biopolymer family that is obtained using microorganisms. The *Azotobacter chroococcum 7B* strain was used for the biosynthesis of PHAs. The polymers synthesized by this producing bacterial strain have low polydispersity, a widespread spectrum of the 3-HB/3-HV monomer ratio in the PHB chain, and MM variability. By varying these parameters during the synthesis of copolymers, their properties, such as Young’s modulus, biodegradation rate, and biocompatibility, can also be varied.

The physicochemical parameters of PHAs are changed in different ways during degradation in solutions of PBS and PBS with lipase. The initial mass of polymer films of various copolymers remained virtually unchanged during the entire 183 days of the experiment (Figure 1). The decrease in film mass during the first week can be explained by the dissolution of water-soluble oligomers and monomers during their desorption from the film into aqueous media.

The change in MM is an important characteristic describing the degradation of polymers. As mentioned earlier, using different methods of growing polymer producers, it is possible to vary the molecular mass of the biosynthesized product. This procedure allows development of devices from PHAs with a programmable rate of degradation. The articles presented recently [1,29,44,45] described the hydrolytic degradation of polymers in the presence of various agents. But, in these works, explanations of reaction mechanisms were not presented. This work is one of the first which to attempt to compare the different behavior of biopolyesters in PBS and in the medium containing pancreatic lipase as one of the most important components involved in the decomposition of PHA in the human body. The study of molecular mass changes during degradation allows us to elucidate this mechanism of the process.

Surprisingly, an increase in molecular mass was observed for PHBV 17.6% 1190 and PHBV 5.9% 819 (Figure 2A,B). The initial increase in molecular mass is associated with leaching of the low molecular mass polymer fraction. This means that, in the manufacture of a polymer film, in its volume, and on the surface, there are short-chain polymer residues that dissolve relatively well in water. When studying the molecular mass of polymers prior to incubation in experimental solutions, short-chain residues will affect the average molecular mass. Upon further placement of the polymer films in aqueous solutions, these residues dissolve and are washed out of the polymer film. This leads to a slight increase in molar mass during subsequent measurement. The hydrolysis of the amorphous polymer component also occurs, which is located on the surface of the film, because, according to the literature, the amorphous component decomposes 20 times faster than crystalline [47]. The subsequent strong decrease in MM testified to the fact that the degradation of these polymers was carried out not only on the surface of the polymer film but partially in volume. The absence of mass reduction of polymer films and Young’s modulus change (due to the process of secondary crystallization of polymers, and also, probably, to the elution of the amorphous component) (Figure 1 and Figure 7) prompt the conclusion that our polymers will retain their structural and mechanical properties over time and perform their function. Thus, the synthesized polymers can find application in the field of bone tissue engineering for the manufacture of implants. The use of calculation models (Table 2) suggested that, in the case of lipase addition, the process of autocatalysis began to effect the rate of degradation, in contrast to hydrolysis in phosphate buffer. From the Table 2 data, it could be concluded that all copolymers and homopolymer, PHB, are better described by a notautocatalytic degradation model, since the values of R^2^ correspond to it. The similar behavior of partially crystalline polymers was presented in the study of Han et al. [45]. In addition, when constructing models and calculating the slope of the curves, a transition point of 5.9% of 3-HV in the copolymer was obtained. Probably, the presence of a more branched radical in the copolymer leads to the fact that it is more difficult for water molecules to hydrolyze the ester bonds. And these steric hindrances begin to affect the entire polymer precisely when the content of 3-HB is greater than 5.9%. In the future, taking this factor into account can help to more correctly calculate the rate of degradation of medical materials based on PHB and its homologues, PHA.

According to published data, the degree of crystallinity of PHA is quite high and varies between 40–70% [42,50]. These data are confirmed by our studies. The degree of crystallinity of the homopolymer was 63%, and the PHBV copolymer 17.6% 1340 was almost two times less −34%. Differences in the degree of crystallinity of polymer films during degradation in PBS and lipase solutions were also demonstrated for the first time. A wave-like evolution in the degree of crystallinity during degradation in PBS was proposed recently [44,49]. In these articles, it was assumed that the changes would not go linearly. We showed that the degree of crystallinity during hydrolysis will have a wave-shape change. All of this is due to crystallization and recrystallization processes. This means the following. During degradation, the amorphous component decomposes faster than the crystalline component 20 times [47]. This leads to an increase in the degree of crystallinity in the polymer. However, hydrolysis occurs non-directionally, that is, the crystalline part also decomposes. Upon decomposition of the crystalline part, an amorphous component is formed—weaving—that leads to a decrease in the degree of crystallinity. Further, the decomposition of the amorphous component or its secondary crystallization occurs, which again leads to an increase in the degree of crystallinity. All of this will have a wave-like appearance, which we have shown in our work. It should be noted that the PHB-4MV copolymer with a 4-MV content of 0.6%, in its physical properties, in particular, crystallinity, is between the PHBV with a content of 5.9% and 9%. It seems that 4-MV makes significant conformational changes in the three-dimensional structure of the copolymer. At the same time, the addition of PEG to the composition of the copolymer does not explicitly affect the polymer structure—the degree of crystallinity of PHBV-PEG is between the copolymer with a 3-HV content of 2.5% to 5.9%. The 3-HV molar content in PHB-PEG is 4.86%. Preliminary, we observed three types of morphological changes in ultrathin PHB films under enzymatic degradation: the emergence of new lamellar structures, fragmentation of lamellar structures, and the disappearance of lamellar structures (Appendix A). However, the degree of crystallinity of the polymers changed differently (from PBS) in the presence of a lipase. But, more important is the result that describes the changes in the behavior of the degree of crystallinity during degradation, depending on the content of HV included in the composition (Figure 9).

Based on the obtained data, it can be concluded that when HV comonomer is included in the molecular chain less than 5.9% of PHBV, there is the increase in degree of crystallinity for the films during degradation. As the percentage of HV increases, the reverse process occurs—the decrease in the degree of crystallinity. This is probably due to the influence of HV groups while the small HV content does not prevent the crystallization of newly formed chains of PHB. Probably, HV comonomer creates steric hindrances for folding into the perfect crystals, which leads to a decrease in crystallinity. Owing to its importance, the above phenomenon requires further study since it can affect the change in mechanical and transport (barrier) properties during degradation. In addition, this phenomenon must be taken into account in the future when predicting the properties of products based on PHA that are contacted with tissues during applications as implants or food packaging.

## 5. Conclusions

Thus, a comprehensive study of the changes in the physicochemical properties of PHB and its copolymers with 3-hydroxyvalerate with different monomer content during the long-term enzymatic and hydrolytic biodegradation under in vitro model conditions was performed. In vitro biodegradation of polymers was studied in PBS and in PBS in the presence of pancreatic lipase at 37 °C for 183 days. An insignificant drop in the mass of the polymers was revealed. However, the change in molecular mass was more significant: a molecular mass decreases up to 80% was found in PHB 1095 kDa in both solutions. In addition, it was shown that, during the enzymatic degradation, the effect of autocatalysis was observed, which was not observed during the hydrolytic degradation. It was also found that Young’s modulus of the copolymers was lower than that of the homopolymer. The incubation of polymer films in both solutions led to an increase in Young’s modulus by more 2-fold. The changes in the degree of crystallinity of the polymers were wave-shape. The films based on biopolymers became more hydrophilic during biodegradation. It was found that the value of ~5.7–5.9% of HV content in the PHBV copolymer was a transition point for changes in the structural and mechanical properties of the PHBV during its degradation.

## Figures and Tables

**Figure 1 polymers-12-00728-f001:**
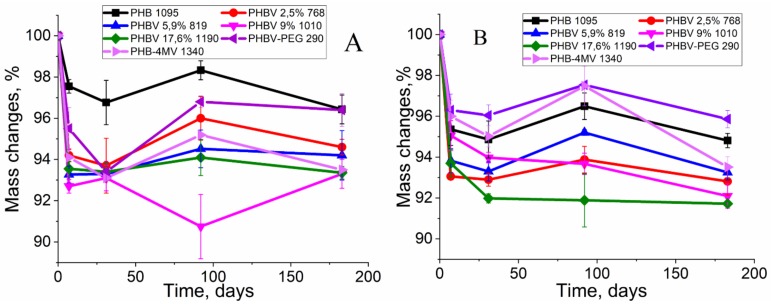
The change in the mass of poly(3-hydroxybutyrate) (PHB), poly(3-hydroxybutyrate-co-3-hydroxyvalerate) (PHBV) (with a different molar content of hydroxyvalerate (HV)), PHB-4 methylvalerate (MV), PHBV-polyethylene glycol (PEG) films during degradation for 6 months in phosphate buffer solution (**A**) and in the same phosphate buffer solution supplemented with the pancreatic lipase (0.25 mg/mL) (**B**).

**Figure 2 polymers-12-00728-f002:**
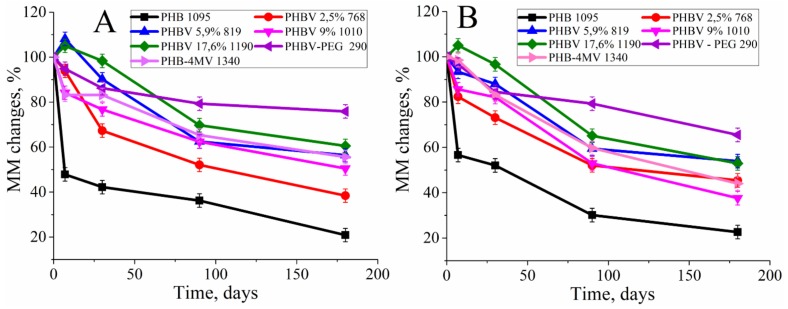
The change in the molecular mass of PHB, PHBV (with a different molar content of HV), PHB-4MV, PHBV-PEG films during hydrolytic (**A**) and enzymatic degradation (**B**) for 6 months.

**Figure 3 polymers-12-00728-f003:**
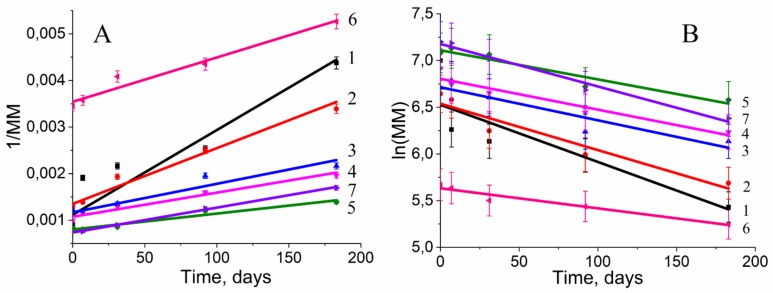
Model of not autocatalytic degradation (**A**) and autocatalytic degradation (**B**), constructed on the basis of the results of changes in the molecular mass of polyhydroxyalkanoates (PHA) in the process of hydrolytic degradation. 1 – PHB 1095, 2 – PHBV 2.5% 768, 3 – PHBV 5.9% 819, 4 – PHBV 9% 1010, 5 – PHBV 17.6% 1190.

**Figure 4 polymers-12-00728-f004:**
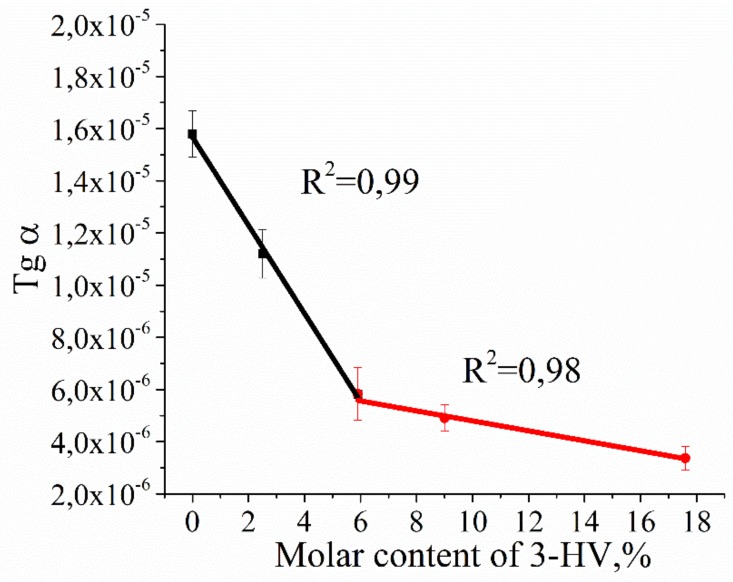
The graph of the slope angle on HV content for the not autocatalytic model of PHBV hydrolytic degradation.

**Figure 5 polymers-12-00728-f005:**
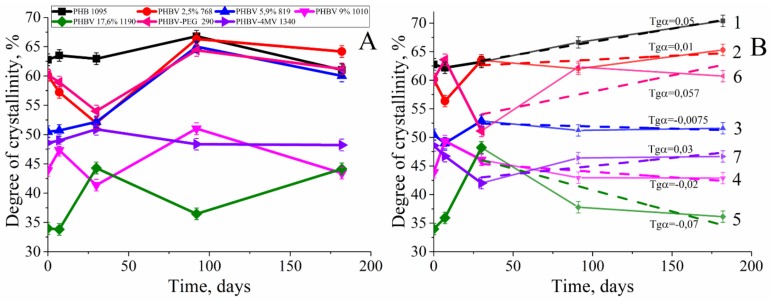
The change in the crystallinity of PHB, PHBV (with a different molar content of HV), PHB-4MV, PHBV-PEG films during hydrolytic (**A**) and enzymatic degradation (**B**) for 6 months: 1 – PHB 1095, 2 – PHBV 2.5% 768, 3 – PHBV 5.9% 819, 4 – PHBV 9% 1010, 5 – PHBV 17.6% 1190, 6 – PHBV-PEG 290, 7 – PHB-4MV 1340.

**Figure 6 polymers-12-00728-f006:**
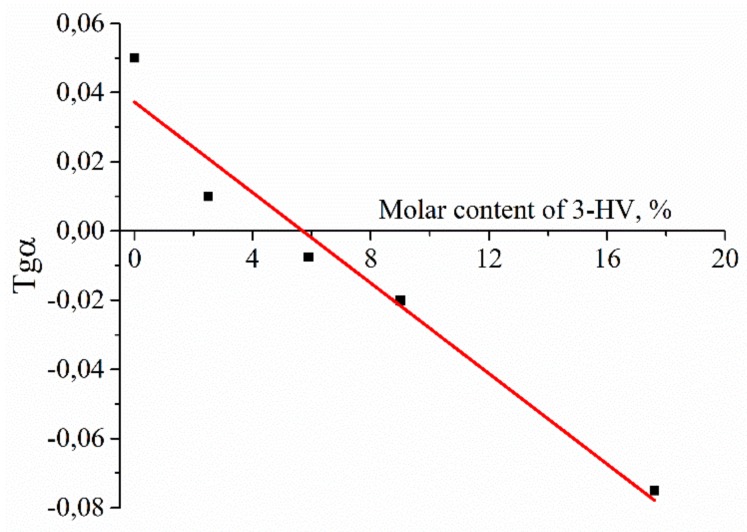
The effect of the 3-HV molar content in the PHBV chain on the crystallinity degree of the copolymers in the degradation process in the lipase solution for 6 months, in the coordinates of the slope of the incubation time.

**Figure 7 polymers-12-00728-f007:**
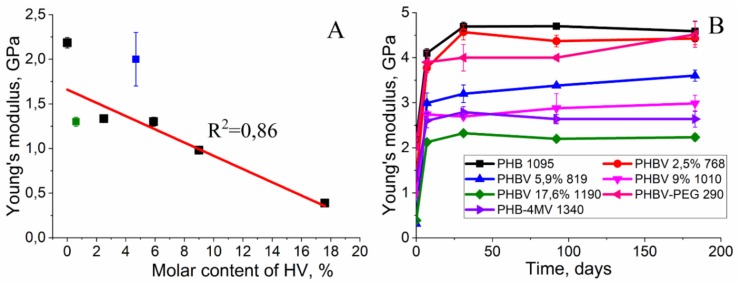
(**A**) Graphs of the dependence of the young’s modulus on the content of 3-HV. The green square corresponds to the PHB-4MV 1340 copolymer, and the blue square corresponds to the PHBV-PEG 290 copolymer. (**B**) Changing Young’s modulus of the polymers in the process of enzymatic degradation.

**Figure 8 polymers-12-00728-f008:**
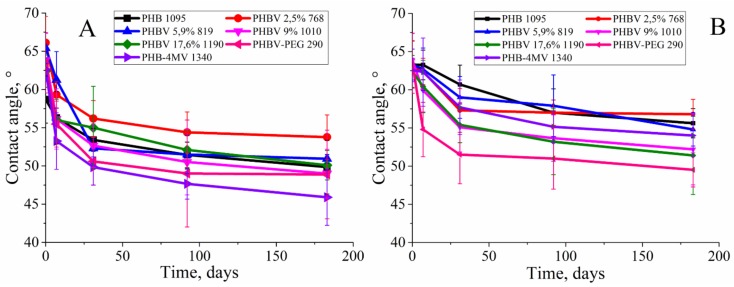
The changes of the contact angle at the water/film boundary of a PHA with a different HV molar content in the PHB chain during enzymatic degradation (**A**) and hydrolytic degradation (**B**) for 6 months.

**Figure 9 polymers-12-00728-f009:**
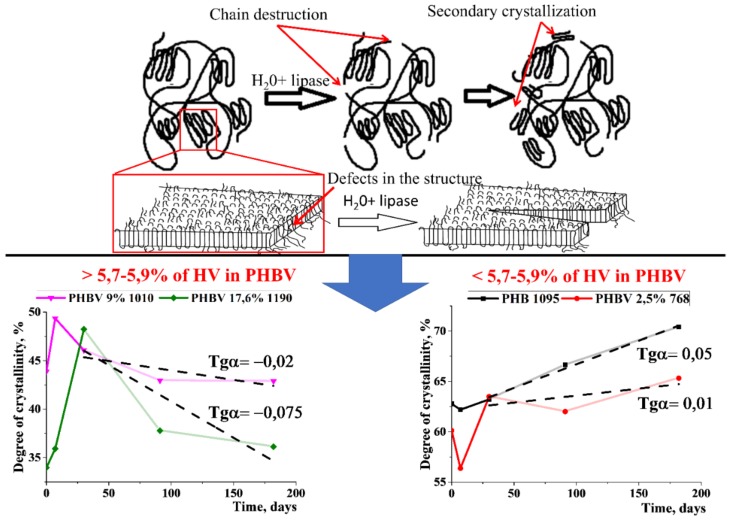
The model of in vitro degradation of PHB and PHBV. It was found that in the process of degradation, the degree of crystallinity increases if the content of 3-HV in the copolymer is less than 5.9% and decreases if it is more than 5.9%.

**Table 1 polymers-12-00728-t001:** The list of polymers used in the work. HV = hydroxyvalerate; PEG = polyethylene glycol; PHB = poly(3-hydroxybutyrate); PHBV = poly(3-hydroxybutyrate-co-3-hydroxyvalerate); MV = methylvalerate.

Substrate	Molecular Mass, kDa	The Content of 3-HV/(3-H4MV)/(PEG) in the Copolymer, %
PHB 1095	1095	0
PHBV 2.5% 768	768	2.5
PHBV 5.9% 819	819	9.0
PHBV 9% 1010	1010	17.6
PHBV 17.6% 1190	1190	5.9
PHBV-PEG 290	290	4.69% 0.15% (PEG)
PHB-4MV 1340	1340	0.60 (3-H4MV)

**Table 2 polymers-12-00728-t002:** Correlation coefficients of a not-autocatalytic and autocatalytic degradation models.

Sample	Hydrolytic Degradation		Enzymatic Degradation	
	R^2^ (Not-Autocatalytic Model)	R^2^ (Autocatalytic Model)	R^2^ (Not -Autocatalytic Model)	R^2^ (Autocatalytic Model)
PHB 1095	0.88	0.68	0.95	0.83
PHBV 2.5% 768	0.99	0.93	0.93	0.88
PHBV 5.9% 819	0.96	0.89	0.92	0.90
PHBV 9% 1010	0.96	0.90	0.99	0.97
PHBV 17.6% 1190	0.97	0.93	0.96	0.95
PHBV-PEG 290	0.92	0.83	0.99	0.98
PHB-4MV 1340	0.97	0.91	0.99	0.98

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
