# Peer review of "Comparative Structure-Property Characterization of Poly(3-Hydroxybutyrate-Co-3-Hydroxyvalerate)s Films under Hydrolytic and Enzymatic Degradation: Finding a Transition Point in 3-Hydroxyvalerate Content"

_polymers, 2020, doi:10.3390/polym12030728_

Round 1
Reviewer 1 Report
In general, the paper is well written and contains interesting data that is worth publishing. Sometimes authors are attempting to gain more information from the rather erratic behavior than it is possible in my opinion.
The abstract should contain more quantitative results and less introducing information.
Please, briefly describe GPC method.
L126-127 There is a type error, please check.
Fig1 The weight change occurred during the first week only. It thus could be caused simply by leaching out some low MW fraction. Please, comment.
Fig2 The change in MW is indeed interesting
L189 type error, please check
Table1 The presentation of the 1 is of course problematic. It is obvious that it is difficult to decide about the mechanism based on the current data.
The manuscript can be published after minor changes.
Reviewer 2 Report
polymers-738531
The authors present in their resubmitted work entitled ‘Comparative structure-property characterization of poly(3-hydroxybutyrate-со-3-hydroxyvalerate)s films under hydrolytic and enzymatic degradation: finding a transition point in 3-hydroxyvalerate content’ a long-term study on the effects of various concentrations of HV in the PHBV films with respect to polymer structure and characteristics, subjecting the polymer preparations to hydrolytic and enzymatic cleavage conditions.
The work is structured well, and processes generating the various data discussed are by and large clearly presented (see below). The proposed concentration limit for HV that seems to drastically change polymer behaviours is delineated in straight forward manner. Some aspects need nevertheless attention before eventually publishing this work.
Specific comments:
- A table is needed at some point, best at the beginning of the results section that lists all realised polymers with their starting characteristics in form of a table, in which also compositions are clearly listed.
- The understandable initial weight loss is followed by an increase of weight for all samples but PHBV 9% 1010 in the enzyme-free and PHBV 17% 1190 in the enzyme-containing case. The exceptions exclude a potential systematic error when weighting that could be attributed to the long time between measurements. This aspect needs some explanation.
- The molecular weight discussion needs more details. Regarding the GPC measurements, what standards have been used for system calibration, which column, which solvent? These data are important to get an idea of the accuracy of the measurements especially in the lower MW regions. More importantly, it needs to be stated which delineable data have been used as Mw as discussed in the text: number average molecular weight (Mn) or the weight average molecular weight (Mw)? Or the peak molecular weight? Additionally, information of the polydispersity would be indicative with respect to a degradation of the polymers. Since all these data are effortlessly available from the GPC measurements, this reviewer suggests to extend the discussion.
- On page 5, line 173; it should say Figure 3A & 3B.
- Figure 5 B needs a revision in the sense that the original plots are better visible together with the tangents.
- The authors should get rid of ‘non-catalytic’, and always explicitly state ‘not autocatalytic’ to facilitate and uniform reading.
- With respect to the various graphical representations: the study would benefit from an easier comparison between the non-enzymatic and enzymatic treatment. e., the graphics within the same figure should be scaled identically, at the moment the ordinate is always longer in B; same percentages should be at the same height. The order of compounds in legends should be kept constant and follow the one introduced in the yet missing table that lists all compounds (comment 1). Figure captions should be organised uniformly (check caption figure 7).
- Always with respect to the various graphical representations: the error bars are barely visible in some graphs, here the settings need to be adjusted to guarantee visibility.
- The abstract is too long and needs shortening.
- The English style is largely OK, but another round of sound proofreading and rephrasing of some very long sentences is necessary.
Given that the authors consider/answer the aforementioned points during revision, the work should become suitable for publishing in polymers.
Reviewer 3 Report
REVISION OF polymers-738531
The present paper performs a Comparative structure-property characterization of poly(3-hydroxybutyrate-со-3-hydroxyvalerate)s films under hydrolytic and enzymatic degradation, in order to find a transition point in 3-hydroxyvalerate content.
The paper deserves some merit, and can be proposed for publication, after the application of some revisions:
FOCUS
The title focused on the finding of a transition point in the 3-HV content. However, despite the result is clear, the argumentation throughout the paper is not clear. More structure would be required.
ABSTRACT
Add GPC in the list of analyses
INTRODUCTION
Continue line 57 explaining the reason
In order to support the novelty of the research, increase the state of the art with previous papers on the effects of HV, the use of PEG or specific copolymers. This is critical.
METHODOLOGY
Define the structures and initial molecular weights of the polymers. This is critical to understand if the results are comparable. In addition, relate the labes of the materials (legends in figures) to the structure.
Clearly define the solve-casting conditions (volume, temperature, stirring, time dissolution, time drying...). This is critical to allow reproducibility in science.
PSB is normally referred as PBS
Specify if the mean molecular weights correspond to the Mp (most-probable MW, i.e. the fashion of the distribution), or if it is MWw or MWn.
Enthalpies are specific and should be represented in lower case.
Explain why only the 1st heating DSC was presented.
Specify how many specimens per sample were analysed for each technique.
Do not use weight, but mass
RESULTS AND DISCUSSION
Despite the present structure is acceptable, the combination of Results and Discussion in the same chapter is more relevant to understand the results. This is just a recommendation.
Correlate with the state of the art the high degrees of crystallinity.
Show the DSC traces, in order to check if there is cold-crystallisation or not, which would affect to the values of teh calculated crystallinity.
Show the stress-strain traces, in order to picture the elastic-plastic behaviour of the polymers.
Add examples of pictures with contact angles.
Detail / expand teh explanation of teh initial increase in Mw
Comemt in line 289 is vague
Analysis of Xc variation needs more discussion and references to support the arguments.
Info on Supl mat S3 should be in the main text
ENGLISH
- Check the entire paper. Hereby a list of lines in whic mistakes have been detected: 24, 277,279.
Round 2
Reviewer 3 Report
The authors have fulfilled most of the proposed revisions and tehrefore can be proposed for publication. Congratulations
Just check the enthalpy. It's a specific variable, and "h", instead of "H" has to be used.